# Signs of neuroaxonal injury in preeclampsia—A case control study

**Malin Andersson[1], Jonatan Oras[1], Sven Egron Thörn [1], Ove Karlsson[1], Peter Kälebo[2], Henrik Zetterberg[3,4,5,6], Kaj Blennow[3,4], Lina Bergman [7,8,9]***

1 Department of Anesthesiology and Intensive Care, Institute of Clinical Sciences, Sahlgrenska Academy, University of Gothenburg, Gothenburg, Sweden, 2 Department of Radiology, Institute of Clinical Sciences, Sahlgrenska Academy, University of Gothenburg, Gothenburg, Sweden, 3 Department of Psychiatry and Neurochemistry, Institute of Neuroscience and Physiology, Sahlgrenska Academy, University of Gothenburg, Mölndal, Sweden, 4 Clinical Neurochemistry Laboratory, Sahlgrenska University Hospital, Mölndal, Sweden, 5 Department of Neurodegenerative Disease, UCL Institute of Neurology, Queen Square, London, United Kingdom, 6 UK Dementia Research Institute, London, United Kingdom, 7 Department of Obstetrics and Gynecology, Institute of Clinical Sciences, Sahlgrenska Academy, University of Gothenburg, Gothenburg, Sweden, 8 Department of Women's and Children's Health, Uppsala University, Uppsala, Sweden, 9 Department of Obstetrics and Gynecology, Stellenbosch University, Cape Town, South Africa

* lina.bergman.2@gu.se

**Data Availability Statement:** Data cannot be shared publicly because of a small population that could be identified. Data are available after approval from the Ethics Committee (contact via national ethics board, 004610-475 08 00) or

## Abstract

### Background

Cerebral injury is a common cause of maternal mortality due to preeclampsia and is challenging to predict and diagnose. In addition, there are associations between previous preeclampsia and stroke, dementia and epilepsy later in life. The cerebral biomarkers S100B, neuron specific enolase, (NSE), tau protein and neurofilament light chain (NfL) have proven useful as predictors and diagnostic tools in other neurological disorders. This case-control study sought to determine whether cerebral biomarkers were increased in cerebrospinal fluid (CSF) as a marker of cerebral origin and potential cerebral injury in preeclampsia and if concentrations in CSF correlated to concentrations in plasma.

### Methods

CSF and blood at delivery from 15 women with preeclampsia and 15 women with normal pregnancies were analysed for the cerebral biomarkers S100B, NSE, tau protein and NfL by Simoa and ELISA based methods. MRI brain was performed after delivery and for women with preeclampsia also at six months postpartum.

### Results

Women with preeclampsia demonstrated increased CSF- and plasma concentrations of NfL and these concentrations correlated to each other. CSF concentrations of NSE and tau were decreased in preeclampsia and there were no differences in plasma concentrations of NSE and tau between groups. For S100B, serum concentrations in preeclampsia were increased but there was no difference in CSF concentrations of S100B between women with preeclampsia and normal pregnancy.

registrator@etikprovning.se for researchers who meet the criteria for access to confidential data.

**Funding:** HZ is a Wallenberg Scholar supported by grants from the Swedish Research Council (#2018-02532), the European Research Council (#681712), the Swedish state under the agreement between the Swedish government and the County Councils, the ALFagreement (#ALFGBG-720931), the Alzheimer Drug Discovery Foundation (ADDF), USA (#201809-2016862), and the UK Dementia Research Institute at UCL. KB is supported by the Swedish Research Council (#2017-00915), the Alzheimer Drug Discovery Foundation (ADDF), USA (#RDAPB-201809-2016615), the Swedish Alzheimer Foundation (#AF-742881), Hjärnfonden, Sweden (#FO2017-0243), the Swedish state under the agreement between the Swedish government and the County Councils, the ALF-agreement (#ALFGBG-715986), and European Union Joint Program for Neurodegenerative Disorders (JPND2019-466-236). LB is supported by the Swedish Society for Medical Research (SSMF).

**Competing interests:** HZ has served at scientific advisory boards for Denali, Roche Diagnostics, Wave, Samumed and CogRx, has given lectures in symposia sponsored by Fujirebio, Alzecure and Biogen, and is a co-founder of Brain Biomarker Solutions in Gothenburg AB (BBS), which is a part of the GU Ventures Incubator Program. KB has served as a consultant or at advisory boards for Abcam, Axon, Biogen, Lilly, MagQu, Novartis and Roche Diagnostics, and is a co-founder of Brain Biomarker Solutions in Gothenburg AB (BBS), which is a part of the GU Ventures Incubator Program. The remaining authors report no conflict of interest.

## Conclusion

NfL emerges as a promising circulating cerebral biomarker in preeclampsia and increased CSF concentrations point to a neuroaxonal injury in preeclampsia, even in the absence of clinically evident neurological complications.

## Introduction

Cerebral complications to preeclampsia such as eclampsia and cerebral hemorrhage are among the most common causes of direct maternal death [1]. In addition to short term complications, women with preeclampsia run an increased risk of developing dementia, stroke and seizure disorders later in life [2–4].

Prediction of eclampsia and cerebral edema in the acute phase is challenging and clinical symptoms such as headache and visual disturbances demonstrate poor prognostic accuracy [5]. Studies in the field are few and limited to animal studies and retrospective studies focused on women who already developed eclampsia [6, 7].

There is a debate about if preeclampsia contributes to later dementia, stroke and seizure disorders or if it is merely a stress test for cerebrovascular disease [8]. Thus, there is a need for an increased understanding of the direct cerebral effects of preeclampsia and there is also a need for an objective biomarker reflecting the degree of cerebral insult, preferably before complications such as eclampsia occur.

Our group and others have showed that peripheral concentrations of the cerebral biomarkers S100B, neuron specific enolase (NSE), tau protein and neurofilament light chain (NfL) are increased in preeclampsia both before onset, during disease and postpartum [9–16]. S100B was the first cerebral biomarker that was reported to be increased in plasma in preeclampsia with severe features by our group and others and in addition, women with visual disturbances demonstrated an increased plasma concentration of S100B [11, 14, 17]. Following these initial reports, there were further studies reporting increased peripheral concentrations also of NSE, NfL and tau, reflecting neurons and axons respectively [12, 16]. There is a gap in knowledge whether these biomarkers truly reflect a cerebral injury in preeclampsia or if they originate from other tissues since preeclampsia is a generalized endothelial disease and the biomarkers are not unique to the central nervous system [18].

Therefore, we sought to determine whether these cerebral biomarkers were also increased in cerebrospinal fluid (CSF) as a marker of cerebral origin and potential cerebral injury in preeclampsia and if there was a correlation to plasma concentrations in order to validate the cerebral origin of circulating concentrations. In addition, results were related to intracerebral findings on magnetic resonance imaging.

## Materials and methods

This study was approved by the regional ethics committee at Gothenburg University (Dnr: 932–16) and all women were included after written and verbal informed consent.

### Population

Women with preeclampsia and women with normal pregnancies over 18 years of age that were delivered by cesarian section were approached for inclusion in the study. All women were recruited at the Department of Obstetrics and Gynecology, Sahlgrenska University

Hospital Östra, Sweden, from January 2017 until May 2018. We included 15 women with pre-eclampsia with severe features according to clinical criteria from the International Society for the study of Hypertension in pregnancy [19] and 15 women with normal pregnancies preoperatively to cesarean section. Demographics were recorded from the medical charts.

## Sample collection

Plasma, serum and CSF were collected at time of caesarian section with spinal anastesia. After collection, the blood and CSF samples were kept in room temperature for no longer than half an hour (half an hour for serum) before being centrifuged for 10 minutes at 2,800 g whereafter they were frozen and stored in -80 degrees C until analysis.

## Variables

Exposure was defined by a diagnosis of preeclampsia with severe features at the discretion of the clinician and confirmed by ICD code O14.1 or O14.2 in the medical chart. Outcomes were plasma, serum and CSF concentrations of cerebral biomarkers and cerebral edema or white matter lesions on magnetic resonance imaging (MRI). Confounders were defined by a factor that had the potential to influence both the exposure and the outcome and in this study, BMI and parity were selected as confounders and registered from the woman's medical chart.

## Biomarker assays

The concentrations of tau and NfL in plasma were measured using the NF-light and Total Tau 2.0 kits using the Simoa platform (Quanterix, Billerica, MA) as previously described in detail [20]. Calibrators were run in duplicates, while samples were run in singlicates with a 4 fold dilution. Two quality control (QC) samples from plasma were run in duplicates in the beginning and the end of each run. For NfL, between-run precision was 6.0% at 8.5 pg/mL and 5.1% at 121 pg/mL, while for T-tau, between-run precision was 7.3% at 32.2 pg/mL and 7.0% at 7.5 pg/mL. Plasma NSE, serum S100B and CSF S100B and NSE concentrations were measured as singlicates on the cobas Elecsys platform, according to the manufacturer's recommendations. CSF T-tau was measured using the INNOTEST enzyme-linked immunosorbent assay (ELISA, Fujirebio Europe, Ghent, Belgium), while CSF NfL was measured using an in-house ELISA as described previously in detail [21]. Elecsys- and ELISA-based methods were chosen for biomarkers for which Simoa ultrasensitivity was not needed. Between-run precision was <10% for all these assays. All measurements of CSF and plasma/serum concentrations were performed in one round of experiments using a single batch of reagents for each assay by board-certified laboratory technicians who were blinded to clinical data.

## Magnetic resonance imaging of the brain

MRI brain was performed twice, first within 48 hours from delivery and for women with preeclampsia, a second examination was performed six months postpartum.

A Philips Ingenia 1.5 T scanner was used. The MRI protocol included standard clinical sequences: T1 weighted TSE sagittal, 3D Brain View FLAIR with reconstructions in 3 planes, T2 weighted TSE axial, Diffusion weighted imaging (DWI) axial, with b values 0 and 1000 s/mm2, susceptibility weighted sequence (SWIp) to detect possible microbleeds and a 3D TOF angiography of intracranial arteries. White matter lesions (WMLs) were considered present if hyperintense on FLAIR and T2 weighted images and not hypointense on a FLAIR or a T1-weighted image. They were registered as subcortical or periventricular depending on location where periventricular WMLs were defined as being located within 3 mm from the

ventricular surface. Cerebral edema of vasogenic origin (no restricted diffusion) or cytotoxic origin (restricted diffusion) was graded as mild or severe and by localization. Due to small numbers in this study, data was presented as presence or not of subcortical–and periventricular WMLs irrespective of size.

## Statistics

Demographic and clinical characteristics were presented as means with standard deviations (SD) and percentages as appropriate and were compared between women with preeclampsia and women with normal pregnancies by use of Student's t-test and chi-square tests. CSF and circulating concentrations of cerebral biomarkers were presented as medians with interquartile range (IQR) and compared between women with preeclampsia and women with normal pregnancies by use of Mann–Whitney U-test and adjusted for confounders in a logistic regression analysis. The albumin quotient was calculated by CSF concentrations divided by circulating concentrations. Associations between CSF and circulating concentrations of the four biomarkers were analyzed using a proportional odds model tailored to countinuous outcome variables described in detail elsewhere [22]. The model is less sensitive to the assumptions of an ordinary linear model and the results are less influenced by extreme outcome observations. The models were adjusted for the potential confounding effects of BMI and parity and a group by biomarker interaction was included to allow for different associations within the two groups. Model fit was assessed visually by comparing the probability scale residuals from the models with the quantiles from a uniform distribution [23]. Data and statistical analyses were performed using SPSS version 26.0 (SPSS; PASW statistics) for MAC software package and R version 3.6.1 using the add-on package rms [24].

## Sample size

Since concentrations of cerebral biomarkers in CSF in preeclampsia have not been evaluated before, sample size was calculated from previous publications of cerebral biomarkers in plasma in preeclampsia. For NfL, an SD of 6.9 and medians of 8.4 and 22 ng/L respectively, a sample size of 4 women in each group was needed. For NSE, the numbers needed were 10 in each group with an SD of 1.1 and medians of 3.1 and 4.5 ug/L respectively. For S100B, the numbers needed were 16 in each group with an SD of 0.03 and a median of 0.04 and 0.07 ug/L respectively. For tau, the numbers needed were 1100 in each group with an SD of 4.2 and a median of 3.8 and 4.3 respectively [9]. Since the CSF concentrations in preeclampsia were not known from before and for practical reasons, the sample size for this pilot study was set to 15 women in each group.

## Results

### Participants

Out of the 30 women approached for inclusion and included, all women underwent blood sampling and biomarker analyses. 12 women with preeclampsia and 13 women with normal pregnancies underwent MRI at inclusion and in the group of women with preeclampsia, 8 women underwent a follow up MRI examination at one year postpartum.

### Background characteristics

Maternal characteristics and pregnancy outcomes of women with preeclampsia and women with normal pregnancies are presented in Table 1. Women with preeclampsia had a higher body mass index (BMI), were more often nulliparous, had a shorter gestational length at

**Table 1. Background characteristics of the population.**

| | Preeclampsia (n = 15) | Normal pregnancy (n = 15) | p-value |
|---|---|---|---|
| Maternal age, years | 32.5 (5.8) | 31.9 (3.7) | ns |
| BMI, in first trimester, $kg/m^2$ | 27.5 (3.8) | 22.9 (3.1) | <0.01 |
| Smoker, n (%) | 1(7) | 1 (7) | ns |
| Nulliparous, n (%) | 12 (80) | 5 (33) | <0.05 |
| Chronic hypertension, n (%) | 1 (7) | 0 (0) | ns |
| **At inclusion** | | | |
| Systolic blood pressure, mm Hg | 154.5 (11.0) | 116.9 (11.0) | <0.001 |
| Diastolic blood pressure, mm Hg | 94.8 (11.4) | 78.8 (8.5) | <0.001 |
| Mean arterial blood pressure, mm Hg | 114.7 (9.7) | 91.5 (9.0) | <0.001 |
| Blood pressure medication, n (%) | | | <0.001 |
| *No medication* | 1(7) | 15(100) | |
| *Oral medication only* | 8 (53) | 0 (0) | |
| *Iv medication or oral and iv medication* | 6 (40) | 0 (0) | |
| Neurological symptoms | | | |
| Headache, n (%) | 9 (60) | 0 (0) | |
| Visual disturbances, n (%) | 5 (33) | 0 (0) | |
| Increased tendon reflexes, n (%) | 2 (13) | 0 (0) | |
| HELLP syndrome, n (%) | 1 (7) | 0 (0) | |
| Increased liver enzymes, n (%) | 5 (33) | 0 (0) | |
| Trombocytopenia, n (%) | 3 (20) | 0 (0) | |
| Epigastric pain, n (%) | 1 (7) | 0 (0) | |
| $MgSO_4$ treatment, n (%) | 3 (20) | 0 (0) | |
| Gestational length at delivery, days | 243.0 (28.3) | 273.5 (35) | <0.001 |
| Birth weight, g | 2200 (991) | 3430 (327) | <0.001 |

Values are presented as means (SD), n, %, Student's t-test and Chi Square test.

BMI, body mass index; HELLP, Hemolysis Elevated Liver Enzymes and low Platelets; MgSO₄, Magnesium Sulphate; ns, non significant.

delivery and delivered an infant of lower birthweight compared to women with normal pregnancies.

At inclusion in the study when biological samples were obtained, 14 women (93%) were treated with antihypertensive medication. Five (36%) of the women with preeclampsia demonstrated increased plasma concentrations of liver enzymes. Three women (20%) had trombocytopenia and out of these, one (13%) had HELLP syndrome. Three women (20%) were treated with $MgSO_4$ at time of inclusion. A majority of women with preeclampsia experienced neurological symptoms where nine (60%) had headache, three (13%) had increased tendon reflexes and five (33%) had visual disturbances. None of the women with preeclampsia experienced eclampsia or focal neurological impairment.

## Circulating- and CSF concentrations of cerebral biomarkers and albumin quotient

Concentrations of cerebral biomarkers and albumin quotient are presented in Table 2. Women with preeclampsia demonstrated increased CSF concentrations of NfL compared to women with normal pregnancies (396 *vs* 336 pg/ml, p<0.01). CSF concentrations of NSE and tau were decreased in women with preeclampsia compared to women with normal

**Table 2. CSF- and plasma concentrations of cerebral biomarkers in preeclampsia and normal pregnancy.**

| | Preeclampsia (n = 15) | Normal pregnancy (n = 15) | p-value |
|---|---|---|---|
| **CSF concentrations** | | | |
| S100B (ug/L) | 0.97 (0.77–1.16) | 0.96 (0.75–1.43) | ns |
| NSE (ug/L) | 6.16 (4.16–8.39)* | 7.56 (6.88–9.67)* | <0.05 |
| tau (pg/ml) | 228 (180–267) | 315 (247–345) | <0.05 |
| NfL (pg/ml) | 396 (357–447) | 336 (292–383) | <0.01 |
| **Circulating concentrations** | | | |
| S100B (ug/L) | 0.08 (0.06-0-10) | 0.05 (0.04–0.06) | <0.01 |
| NSE (ug/L) | 10.05 (8.66–15.22) | 11.44 (8.94–12.04) | ns |
| tau (pg/ml) | 3.13 (2.81–4.59) | 2.32 (1.88–4.81) | ns |
| NfL (pg/ml) | 9.29 (6.41–14.03) | 5.44 (4.25–7.25) | <0.001 |
| **Albumin ratio** | 3.02 (2.44–4.69) | 2.73 (2.22–3.64) | ns |

Values are presented as medians (interquartile range). Mann Whitney U-test. S100B was measured in serum, the rest of the biomarkers in plasma.

CSF, Cerebrospinal fluid; NSE, Neuron Specific Enolase; NfL, Neurofilament Light Chain; ns, non significant

*Seven women in each group due to hemolysis in the samples.

pregnancies (6.16 *vs* 7.56 ug/L and 228 *vs* 315 pg/ml respectively, p<0.05). There was no difference in CSF concentrations of S100B between women with preeclampsia and women with normal pregnancies.

Women with preeclampsia demonstrated increased serum concentrations of S100B (0.08 *vs* 0.05 ug/L, p<0.01) and plasma concentrations of NfL (9.29 *vs* 5.44 pg/ml, p<0.001) compared to women with normal pregnancies. Plasma concentrations of NSE and tau were not significantly different between women with preeclampsia and women with normal pregnancies.

There was no difference in albumin quotient between women with preeclampsia and women with normal pregnancies.

In a logistic regression analyses, adjusting for 1) parity and 2) BMI, only concentrations of NfL in plasma and CSF were more likely to be increased for women with preeclampsia (S1 Table). In further regression analyses investigating the association between CSF- and circulating concentrations of the four biomarkers, adjusting for parity and BMI, only CSF concentrations of NfL were related to plasma concentrations (p = 0.028, Fig 1). The association was mainly present in women with normal pregnancies (OR = 3.89; 95% CI: 1.22–12.4) rather than women with preeclampsia (OR = 2.13, 95% CI: 0.91–5.02) but data did not support a group by marker interaction (S2 Table). For the other biomarkers, no correlation between circulating concentrations and CSF concentrations was found.

## MRI brain at inclusion

MRI findings are presented in Table 3. At inclusion, MRI brain was performed for 12 women with preeclampsia and 13 women with normal pregnancies. Three women with preeclampsia and two women with normal pregnancies cancelled during the MRI as they found the experience claustrophobic.

There was no difference in occurrence of either subcortical or periventricular WMLs between women with preeclampsia and women with normal pregnancies (25% *vs* 23% and 8.3 *vs* 7.6% respectively). One woman with preeclampsia demonstrated features compatible to cerebral edema with cytotoxic signs without any WMLs present (Table 3).

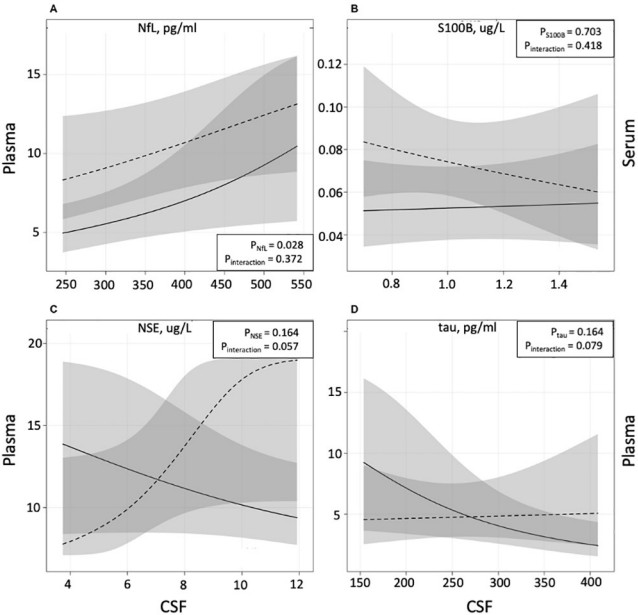

**Fig 1. Associations between CSF- and circulating concentrations of cerebral biomarkers.** Estimated mean values of circulating concentrations as a function of CSF for NfL (A), S100B (B), NSE (C) and tau (D), adjusted for parity and BMI using a cumulative probability model. Preeclampsia is represented by the dashed line and normal pregnancy by the solid line. Grey areas represent pointwise 95% confidence intervals.

## MRI brain at six months postpartum

At six months postpartum, out of the 12 women with preeclampsia that underwent MRI at inclusion, eight women underwent a follow up examination. The remaining four women were lost to follow up.

WMLs persisted in same number and size at six months postpartum in two women. The third woman with WMLs at inclusion was lost to follow up. The cerebral edema in the woman with preeclampsia at inclusion was no longer detectable at the MRI scan at follow up.

## Cerebral biomarkers in relation to MRI brain and clinical findings

The woman with preeclampsia and cerebral edema demonstrated increased CSF concentrations of all cerebral biomarkers S100B, NSE, NfL and tau compared to all other women in the study (1.3 *vs* 0.9 ug/L, 10.6 vs 6.9 ug/L, 591 *vs* 367.5 ng/ml and 284 *vs* 247 ng/ml respectively) and increased circulating concentrations of S100B, NfL and tau compared to all other women in the study (0.14 *vs* 0.06 ug/L, 13.0 *vs* 6.8 ug/L and 234.7 *vs* 2.9 ng/ml respectively) but no

**Table 3. Magnetic Resonance Imaging findings in preeclampsia and normal pregnancy.**

|  | At inclusion | | | At follow up |
| --- | --- | --- | --- | --- |
|  | Preeclampsia (n = 12) | Normal Pregnancy (n = 13) | p-value | Preeclampsia (n = 8) |
| Subcortical WML n(%) | 3 (25) | 3 (23) | ns | 2 (25) |
| Juxtaventricular WML n(%) | 1 (8) | 1 (8) | N/A | 0 (0) |
| Cerebral edema n(%) | 1 (8) | 0 (0) | N/A | 0 (0) |

Values are presented as numbers (%). Mann Whitney U-test.

WML, White matter lesions; ns, non significant N/A non applicable.

statistical analyses were possible due to that only one case presented with cerebral edema. Regarding WMLs, there were no differences in concentrations of cerebral biomarkers at time of delivery or at six months postpartum in women with or without WMLs present.

There was no difference in circulating- or CSF concentrations of cerebral biomarkers in women with preeclampsia and epigastric pain, HELLP, thrombocytopenia, elevated liver enzymes, neurological signs and symptoms or $MgSO_4$ treatment compared with women without these features.

## Discussion

### Key results

In this pilot study of cerebral biomarkers in preeclampsia, the main finding was that CSF concentrations of NfL were increased in preeclampsia. These findings might indicate a neuro-axonal injury in preeclampsia even when clinical and radiological neurological complications are absent. In addition, there was an association between CSF- and plasma concentrations of NfL, in particular in normal pregnancy, also after adjusting for parity and BMI. This finding support the cerebral origin of the increased plasma concentrations of NfL in preeclampsia.

### Interpretation

Our group and others have previously shown that cerebral biomarkers are increased in plasma before onset, during disease and one year after pregnancy in preeclampsia [10, 12, 14, 16]. The results of this study support these findings regarding S100B and NfL but we could not reproduce earlier findings of increased NSE and tau plasma concentrations in preeclampsia.

S100B, NSE, tau and NfL have been evaluated in other neurological disorders before [25–28]. S100B has been used as a marker to rule out intracranial injury in mild traumatic brain injury to avoid unnecessary computer tomography scans [25] and NSE has been used as a prognostic factor in adverse neurological outcome after cardiac arrest [26]. Tau has emerged as a sensitive marker for Alzheimer's disease [27] and NfL is a sensitive marker for axonal injury extensively studied in multiple sclerosis and traumatic brain injury [20, 28–30].

Our results showed that S100B was increased in serum but not in CSF in preeclampsia. This could have different explanations. One reason could be that the signal from S100B produced in other extracerebral tissues such as muscle cells and fat cells are dominating causes to increased serum concentrations in preeclampsia. Another explanation could be that S100B, produced in astrocytic end-feet close to the blood brain barrier (BBB), is secreted in higher amounts into the blood stream due to BBB injury and thus depleted from the CNS. Circulating concentrations of S100B in non pregnant women are reported at around 0.06 ug/L [10].

S100B, tau and NfL demonstrated a 10–100 fold increased concentration in CSF compared to plasma/serum, reinforcing higher abundance in the central nervous system. In contrast, for NSE, plasma concentrations were higher compared to CSF concentrations. The reason for this is not known. We speculate about that the NSE signal in plasma in this study might be derived from other sources such as red blood cells which might also have impacted previous reports of peripheral concentrations of NSE in preeclampsia.

Regarding plasma concentrations of tau, there were no significant differences between groups [12]. In addition, the correlation between plasma concentrations and CSF concentrations of tau in Alzheimer's disease has proven to be weak and other confounding sources of tau have to be considered [27]. These findings are supported in this study where we did not find any correlation between CSF and circulating concentrations of neither tau, NSE or S100B. Thus, alternative extracerebral sources of tau, NSE and S100B in preeclampsia have to be considered.

An unexpected finding was that women with preeclampsia had reduced CSF concentrations of NSE and tau. In the absence of neurodegeneration, extracellular levels of tau are regulated by neuronal activity-dependent release of the protein, [31] which is likely to determine the corresponding CSF concentrations. It is possible that such a mechanism regulates extracellular NSE concentrations as well. Reduced CSF concentrations of tau and NSE in preeclampsia could thus reflect reduced neuronal activity. There are reports about impaired cognitive function and increased risk of dementia after preeclampsia [2, 32] but if this related to decreased CSF concentrations of tau and NSE remains to be proven. It is, however, possible that other mechanisms, potentially related to CSF dynamics, might play a role as well, but then all markers would move in the same direction, which was not the case in our study. In any case, our findings are in line with other publications reporting decreased CSF concentrations of tau in preeclampsia and in placental insufficiency [33, 34]. These findings need to be confirmed in future studies.

Parity is known to affect the brain regarding size and structure [35] and BMI can contribute to an increased secretion of cerebral biomarkers from alternative sources [36] and at the same time they are affecting the risk of exposure (preeclampsia). When adjusting for these confounders, only NfL remained as a discriminatory biomarkers between preeclampsia and normal pregnancy both in CSF and plasma.

In our study, there was no alteration in albumin quotient in preeclampsia *vs* normal pregnancy, supported by previous findings [37]. This quotient is normally used to reflect BBB integrity but it depend largely on the CSF flow where diminished flow will lead to increased CSF concentrations of albumin and hence a picture of an injured BBB even when the BBB is intact [38]. In addition, albumin is a large molecule and there might be other mechanisms of action causing increased permeability over the BBB in preeclampsia such as up-regulation of transporter proteins shown in animal studies [39].

## Strengths and limitations

One of the strengths of this study was the availability of CSF and plasma samples in the same woman where cerebral biomarkers in preeclampsia can be evaluated in CSF and compared to plasma concentrations. Other strengths were the well-defined group of women with preeclampsia and severe features, the availability of MRI examinations to characterize cerebral involvement and the sophisticated analyses of NfL and tau through Simoa technique. The weaknesses of the study include the small sample size and the lack of severe neurological complications in the group of women with preeclampsia.

It is not yet known why tau and NSE were decreased in CSF in preeclampsia on a group level, though these groups are small and findings need to be repeated in larger sample sizes with different phenotypes of preeclampsia. The only woman with cerebral edema demonstrated increased CSF concentrations of all biomarkers. Our speculation would be that a woman with cerebral edema has a larger cerebral insult and thus a potential increased neuronal, glial and axonal injury. Cerebral edema in preeclampsia is thought to be both vasogenic and sometimes cytotoxic. The cytotoxic edema could potentially be non-reversible, causing persistent injury and this could be the cause to the increased concentrations.

In addition, regarding NSE, many of the samples were partly hemolyzed and could not be analyzed for NSE since NSE exists in vast amounts in red blood cells [40]. This could have contributed to that our data differed compared to previous publications. Finally, our findings could not support earlier reports of increased plasma concentrations of tau in preeclampsia. This could be due to a smaller number of women in our study or other phenotypes of preeclampsia compared to other studies since preeclampsia is a heterogenous disorder.

## Conclusion

We have shown that NfL has potential to be a promising cerebral biomarker to reflect possible neuroaxonal injury in preeclampsia. Though our findings originate from a small study sample and need to be further investigated both in relation to preeclampsia with clinically evident neurological injury such as eclampsia and cerebral edema as well as more subtle neurological impairment in preeclampsia such as neurocognitive deficiencies. Should further studies demonstrate increased NfL concentrations in plasma in cases of preeclampsia with neurological impairment, NfL could potentially have a role as prognostic and /or diagnostic biomarker of cerebral involvement in preeclampsia.

## Supporting information

**S1 Table. Concentrations of cerebral biomarkers and association with preeclampsia.**
(PDF)

**S2 Table. Associations of CSF and circulating concentrations of cerebral biomarkers adjusted for BMI and parity and with a group by biomarker interaction for different associations within the two groups.**
(PDF)

**S1 Checklist. STROBE statement—Checklist of items that should be included in reports of case-control studies.**
(DOC)

## Acknowledgments

We would like to thank Erik Lampa, Uppsala Clinical Research Center, for valuable statistical support.

## Author Contributions

**Conceptualization:** Malin Andersson, Sven Egron Thörn, Ove Karlsson.

**Data curation:** Malin Andersson, Lina Bergman.

**Formal analysis:** Peter Kälebo, Henrik Zetterberg, Kaj Blennow, Lina Bergman.

**Funding acquisition:** Sven Egron Thörn, Henrik Zetterberg, Kaj Blennow.

**Investigation:** Malin Andersson.

**Methodology:** Malin Andersson, Jonatan Oras, Sven Egron Thörn, Ove Karlsson, Lina Bergman.

**Project administration:** Malin Andersson.

**Resources:** Peter Kälebo, Henrik Zetterberg, Kaj Blennow, Lina Bergman.

**Software:** Peter Kälebo, Henrik Zetterberg, Kaj Blennow, Lina Bergman.

**Supervision:** Jonatan Oras, Sven Egron Thörn, Lina Bergman.

**Validation:** Peter Kälebo, Henrik Zetterberg, Kaj Blennow, Lina Bergman.

**Writing – original draft:** Malin Andersson, Lina Bergman.

**Writing – review & editing:** Malin Andersson, Jonatan Oras, Sven Egron Thörn, Ove Karlsson, Peter Kälebo, Henrik Zetterberg, Kaj Blennow, Lina Bergman.

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
