## [Decision Letter · Decision Letter 0]

5 Nov 2020

PONE-D-20-29199

Evidence of neuroaxonal injury in preeclampsia – a case control study 

Running head: Neuroaxonal injury in preeclampsia

PLOS ONE

Dear Dr. Bergman,

Thank you for submitting your manuscript to PLOS ONE. After careful consideration, we feel that it has merit but does not fully meet PLOS ONE’s publication criteria as it currently stands. Therefore, we invite you to submit a revised version of the manuscript that addresses the points raised during the review process.

We look forward to receiving your revised manuscript.

Kind regards,

Firas H Kobeissy, PhD

Academic Editor

PLOS ONE

Journal Requirements:

2. We note in your Data Availability statement you have advised "No - some restrictions will apply" and also stated "All relevant data are within the manuscript and its Supporting Information files."

If there are ethical or legal restrictions to sharing your data publicly, please explain these restrictions in detail. Please see our guidelines for more information on what we consider unacceptable restrictions to publicly sharing data: http://journals.plos.org/plosone/s/data-availability#loc-unacceptable-data-access-restrictions. Note that it is not acceptable for the authors to be the sole named individuals responsible for ensuring data access.

3. Please provide additional details regarding participant consent.

In the ethics statement in the Methods and online submission information, please ensure that you have specified what type you obtained (for instance, written or verbal, and if verbal, how it was documented and witnessed).

If your study included minors, state whether you obtained consent from parents or guardians.

If the need for consent was waived by the ethics committee, please include this information.

4. Thank you for stating the following in the Funding Section of your manuscript:

"HZ is a Wallenberg Scholar supported by grants from the Swedish Research Council (#2018-02532), the European Research Council (#681712), the Swedish state under the agreement between the Swedish government and the County Councils, the ALF-agreement (#ALFGBG-720931), the Alzheimer Drug Discovery Foundation (ADDF), USA (#201809-2016862), and the UK Dementia Research Institute at UCL.

KB is supported by the Swedish Research Council (#2017-00915), the Alzheimer Drug Discovery Foundation (ADDF), USA (#RDAPB-201809-2016615), the Swedish Alzheimer Foundation (#AF-742881), Hjärnfonden, Sweden (#FO2017-0243), the Swedish state under the agreement between the Swedish government and the County Councils, the ALF-agreement (#ALFGBG-715986), and European Union Joint Program for Neurodegenerative Disorders (JPND2019-466-236).

LB is supported by the Swedish Society for Medical Research (SSMF)."

We note that you have provided funding information that is not currently declared in your Funding Statement. However, funding information should not appear in the Funding section or other areas of your manuscript. We will only publish funding information present in the Funding Statement section of the online submission form.

 "HZ has served at scientific advisory boards for Denali, Roche Diagnostics, Wave, Samumed and CogRx, has given lectures in symposia sponsored by Fujirebio, Alzecure and Biogen, and is a co-founder of Brain Biomarker Solutions in Gothenburg AB (BBS), which is a part of the GU Ventures Incubator Program. KB has served as a consultant or at advisory boards for Abcam, Axon, Biogen, Lilly, MagQu, Novartis and Roche Diagnostics, and is a co-founder of Brain Biomarker Solutions in Gothenburg AB (BBS), which is a part of the GU Ventures Incubator Program.

The remaining authors report no conflict of interest."

Please also advise if you with to updated your statement of Competing Interests: 'The authors have declared that no competing interests exist'.

5. Please include captions for your Supporting Information files at the end of your manuscript, and update any in-text citations to match accordingly. Please see our Supporting Information guidelines for more information: http://journals.plos.org/plosone/s/supporting-information

Reviewers' comments:

Reviewer's Responses to Questions

**Comments to the Author**

1. Is the manuscript technically sound, and do the data support the conclusions?

Reviewer #1: Yes

Reviewer #2: Partly

2. Has the statistical analysis been performed appropriately and rigorously? 

Reviewer #1: Yes

Reviewer #2: Yes

3. Have the authors made all data underlying the findings in their manuscript fully available?

Reviewer #1: Yes

Reviewer #2: Yes

4. Is the manuscript presented in an intelligible fashion and written in standard English?

Reviewer #1: Yes

Reviewer #2: Yes

5. Review Comments to the Author

Reviewer #1: In the present manuscript titled “Evidence of neuroaxonal injury in preeclampsia – a case control study” authors Dr. Bergman et al., presented a case study work where CSF and plasma from 15 women with preeclampsia and 15 women with normal pregnancies were analyzed for four biomarkers S100B, NSE, tau and NfL by Simoa and ELISA. The main finding is women with preeclampsia demonstrated increased CSF and plasma concentrations of NfL which correlated to each other. However, there was no difference in occurrence of either subcortical or periventricular WMLs between the two groups from the MRI findings. The use of Ultra-sensitive Simoa platform is the strong point in this study to measure the very low levels of the biomarkers. The MRI data provided is also adds strength to the study.

The major issue is that the authors ran the samples in just singlicates which needs an attention from the authors. A replicate of two measurement required when running such tests. The conclusion section is overelaborate without through confirmation or evidence of neuroaxonal injury in preeclampsia.

I recommend that this paper for major revisions and also would like to seek response for below comments for further consideration.

My major comments are below:

Introduction Section:

1. The authors use the term “cerebral biomarkers” (lines 80 and 86) for S100B, NSE, NfL and tau while there are emerging evidence that elevated levels of NfL are a nonspecific marker of neuronal degeneration and do not tell about the underlying cause. I would be cautions of stating them as cerebral biomarkers and rather use the term biomarkers instead of cerebral biomarkers.

2. In the current section the authors briefly mention about their previous work (lines 80-82). I would like the authors discuss or provide more background or rationale on why they choose the above four biomarker to study preeclampsia. This would keep the readers more engaged to the current work and create a flow.

Methods Section:

1. The methods section is confusing. In the lines 120-121 the authors mention that the concentrations of tau and NfL in plasma and CSF were measured using the NF-light and Total Tau 2.0 kit on the Simoa platform. But in the lines 128-130 the CSF Total-tau (T-tau) was measured using the INNOTEST ELISA, and CSF NfL was measured using an in-house ELISA. Is this statement accurate? Why was the CSF T-tau and NfL measured twice using different platforms? Also, what Simoa platform did the authors use (HD1 or HD-X)? Please clarify.

2. For Simoa analysis the calibrators and quality controls (high and low) are run in duplicates. These are in buffer matrix (the calibrators are in some kind of proprietary buffer from Qaunterix) and usually the duplicate measurements have a %CV of <10-20% (which are acceptable range) on Simoa. One of the serious issue I found is that the authors measured the samples in singlicates (as mentioned in lines 123 and 127). The CSF and plasma are true biofluids and I think a duplicate measurement is required at the minimum to assess the experimental integrity and account for the SD (which is not reported). The % CV is cannot established for any of the samples. This specially applies for the ELISA (less sensitive method) tests where duplicate measurements are required.

3. Why did not the authors include a true control for CSF and plasma? Example: Normal (non-pregnant, age and gender matched CSF and plasma). The reason why I ask is this can serve as a background or endogenous levels for the tested biomarkers in the respective matrix (CSF and plasma). Can the authors comment or show data (either performed by them or any other literature) for the baseline levels of these markers in a non-pregnant women? This experiment is critical to corroborate the current results from this study. This is also the reason why I suggested not to use the term cerebral biomarkers as above (Point 1 of the introduction section).

4. The authors state that all measurements were performed in one round of experiments using a single batch of reagents. While this is a good practice to do, this is also a limited practice as it does not tell about the inter-day variations of the biomarkers tested. Did the authors care to perform an inter-day test to determine the assay variations? This specifically happens with Simoa where the controls (low and high) don’t exactly read the same and Quanterix gives a range (not a specific number). Since the sample size is low and on top the authors ran samples in singlicates, a day to day variation would cause certain degree of changes in the measurements.

5. I would be interested to look at the Simoa raw data which can be generated from the batch calibration report. Can I request the authors to share this report?

Results Section:

1. While the results are interesting, can the authors comment on why they see (in CSF) increase for some biomarkers (NfL) and decrease for some biomarkers (NSE and tau) compared to the compared to women with normal pregnancies? Though some explanation is offered in the lines 324-327, I did not find this compelling. What does a decrease in biomarker happen in a clinical sense?

2. Also can authors comment on why the S100B change (increase) was noticed in plasma but not serum? And similarly (no difference) for NSE and tau?

3. Table 2. Why is the units for S100B and NSE denoted in ug/L, while for NfL and tau the units are in pg/mL? Is there a reason why the authors choose to do so? It would be better to see all units in pg/mL, for uniformity in the table.

4. I am not sure how the hemolysis was observed (line 222) only for seven women in each group. Aren’t the same (aliquot/stock) CSF was used all the other biomarker measurement? In that case all the CSF measurements for other biomarkers (for seven women) should have the hemolysis. Please clarify or am I missing something here?. Also for the lines (304-305)

5. I understand that the authors could not perform any stats on the woman with preeclampsia and cerebral edema. But can the authors comment on why the CSF biomarkers (NSE and tau) which was decreased for the preeclampsia group (compared to normal group), is increased for this particular subject? And the same applies for the S100B.

Discussion Section:

1. I truly found the lines 286-288 overstated. The results show CSF concentrations of NfL were increased in preeclampsia but there was no evidence presented to indicate any neuro-axonal injury. Are the authors inferring increased NfL in preeclampsia as direct outcome of neuro-axonal injury while the MRI show normal readings? I would re-write this section.

2. I think the authors should comment and/ or mention in the limitations section on why they could not reproduce earlier findings of increased NSE and tau plasma concentrations in preeclampsia as their older studies (lines 294-297). Similarly the statements made in lines 309-312 can be moved to study limitations rather.

3. A citation is required of the line 298.

4. Can the authors support the claim that NfL extensive marker for multiple sclerosis and traumatic brain injury? Is the claim of increased NfL in preeclampsia in the current study is inferred form the above study? Please explain (this goes with the point 1 above).

5. Lines 308-312 there are too much speculation from the authors. I suggest to re-construct this section to tie it to the current study theme.

6. The strength and limitation sections I think is not well written and requires reconstruction. The above limitation (see above) should be incorporated.

7. Does the MRI examinations reflect a degree of cerebral injury (as stated in lines 347-348)? I don’t see it in Table 3. Please clarify.

Conclusion section:

1. Lines 352-353 is overstated without any direct/tangible evidence for neuroaxonal injury I would refrain from using such sentence without a clear confirmation.

2. Lines 353-354 is confusing to me. What does the author mean here? Please clarify.

3. Lines 357-358 again is overstated. It is not clear how increased CSF concentrations point to a neuroaxonal injury in preeclampsia and how does detection and facilitate treatment? Does the authors mean increased NfL is a diagnostic marker for preeclampsia?

I sincerely request the authors to reconstruct the whole section without these overstatement. I would appreciate any direct evidence for neuroaxonal injury as the authors state here.

Figure-1

The axis titles of the figure are too small. I would suggest the authors to increase the font size for better resolution.

Reviewer #2: This manuscript is a case report study reporting the identification of NFL as biomarker for neuronal injury in preeclampsia. Even though there is a lot of papers reporting on neuronal injury due to preclampsia including papers from the same group, this manuscript is unique in the context of having plasma and CSF samples from the same patient and therefore can correlate the finding between plasma and CSF.

The title is too much strong for the findings and as the authors mentioned in their manuscript that their findings are a pilot study and further analysis are required to confirm their conclusion. the title should be changed

The introduction, materials and methods and result section are very written and presented. however, the discussion need to be readjusted and the finding be discussed more in a global view rather than a point by point discussion. Also. it's clear that the writing style is different from the other parts of the manuscript and re working on the writing style will be more impactful.

6. PLOS authors have the option to publish the peer review history of their article (what does this mean?). If published, this will include your full peer review and any attached files.

Reviewer #1: No

Reviewer #2: No

---

## [Author Response · Author response to Decision Letter 0]

24 Nov 2020

Dear Dr Kobeissy, 

Re: PONE-D-20-29199; Evidence of neuroaxonal injury in preeclampsia – a case control study 

We thank the reviewers for their comments. Here we are pleased to provide our responses. We look forward to further correspondence from PLoS ONE. All changes to the manuscript are marked in yellow.

Kind regards,

Dr Lina Bergman, corresponding author

#Editorial comments

Thank you for pointing this out. We have now made sure that the manuscript follows PLoS ONE:s requirements.

2. We note in your Data Availability statement you have advised "No - some restrictions will apply" and also stated "All relevant data are within the manuscript and its Supporting Information files." If there are ethical or legal restrictions to sharing your data publicly, please explain these restrictions in detail. Please see our guidelines for more information on what we consider unacceptable restrictions to publicly sharing data: http://journals.plos.org/plosone/s/data-availability#loc-unacceptable-data-access-restrictions. Note that it is not acceptable for the authors to be the sole named individuals responsible for ensuring data access. We will update your Data Availability statement to reflect the information you provide in your cover letter.

Ethical approval has been granted to present data on a group level. There are only 15 women in each group that might be identified if data is shared on an individual level. To share data on an individual levels with researchers outside the research group, a new ethical application must be approved.

3. Please provide additional details regarding participant consent.

In the ethics statement in the Methods and online submission information, please ensure that you have specified what type you obtained (for instance, written or verbal, and if verbal, how it was documented and witnessed). If your study included minors, state whether you obtained consent from parents or guardians. If the need for consent was waived by the ethics committee, please include this information.

This is now clarified in the methods section;

“This study was approved by the regional ethics committee at Gothenburg University (Dnr: 932-16) and all women were included after written and verbal informed consent.” 

(page 6, lines 96-97)

4. Thank you for stating the following in the Funding Section of your manuscript:

"HZ is a Wallenberg Scholar supported by grants from the Swedish Research Council (#2018-02532), the European Research Council (#681712), the Swedish state under the agreement between the Swedish government and the County Councils, the ALF-agreement (#ALFGBG-720931), the Alzheimer Drug Discovery Foundation (ADDF), USA (#201809-2016862), and the UK Dementia Research Institute at UCL.

KB is supported by the Swedish Research Council (#2017-00915), the Alzheimer Drug Discovery Foundation (ADDF), USA (#RDAPB-201809-2016615), the Swedish Alzheimer Foundation (#AF-742881), Hjärnfonden, Sweden (#FO2017-0243), the Swedish state under the agreement between the Swedish government and the County Councils, the ALF-agreement (#ALFGBG-715986), and European Union Joint Program for Neurodegenerative Disorders (JPND2019-466-236).

LB is supported by the Swedish Society for Medical Research (SSMF)."

We note that you have provided funding information that is not currently declared in your Funding Statement. However, funding information should not appear in the Funding section or other areas of your manuscript. We will only publish funding information present in the Funding Statement section of the online submission form.

 "HZ has served at scientific advisory boards for Denali, Roche Diagnostics, Wave, Samumed and CogRx, has given lectures in symposia sponsored by Fujirebio, Alzecure and Biogen, and is a co-founder of Brain Biomarker Solutions in Gothenburg AB (BBS), which is a part of the GU Ventures Incubator Program. KB has served as a consultant or at advisory boards for Abcam, Axon, Biogen, Lilly, MagQu, Novartis and Roche Diagnostics, and is a co-founder of Brain Biomarker Solutions in Gothenburg AB (BBS), which is a part of the GU Ventures Incubator Program.

The remaining authors report no conflict of interest."

Thank you for this information. We have looked through the manuscript but do not find a funding statement within the manuscript. We would like the funding statement to read as follows:

"HZ is a Wallenberg Scholar supported by grants from the Swedish Research Council (#2018-02532), the European Research Council (#681712), the Swedish state under the agreement between the Swedish government and the County Councils, the ALF-agreement (#ALFGBG-720931), the Alzheimer Drug Discovery Foundation (ADDF), USA (#201809-2016862), and the UK Dementia Research Institute at UCL.

KB is supported by the Swedish Research Council (#2017-00915), the Alzheimer Drug Discovery Foundation (ADDF), USA (#RDAPB-201809-2016615), the Swedish Alzheimer Foundation (#AF-742881), Hjärnfonden, Sweden (#FO2017-0243), the Swedish state under the agreement between the Swedish government and the County Councils, the ALF-agreement (#ALFGBG-715986), and European Union Joint Program for Neurodegenerative Disorders (JPND2019-466-236).

LB is supported by the Swedish Society for Medical Research (SSMF)."

Please also advise if you with to updated your statement of Competing Interests: 'The authors have declared that no competing interests exist'.

We would like the competing interest statement to read as follows;

 "HZ has served at scientific advisory boards for Denali, Roche Diagnostics, Wave, Samumed and CogRx, has given lectures in symposia sponsored by Fujirebio, Alzecure and Biogen, and is a co-founder of Brain Biomarker Solutions in Gothenburg AB (BBS), which is a part of the GU Ventures Incubator Program. KB has served as a consultant or at advisory boards for Abcam, Axon, Biogen, Lilly, MagQu, Novartis and Roche Diagnostics, and is a co-founder of Brain Biomarker Solutions in Gothenburg AB (BBS), which is a part of the GU Ventures Incubator Program.

The remaining authors report no conflict of interest." 

5. Please include captions for your Supporting Information files at the end of your manuscript, and update any in-text citations to match accordingly. Please see our Supporting Information guidelines for more information: http://journals.plos.org/plosone/s/supporting-information

We have now updated this information accordingly.

Reviewer #1: 

In the present manuscript titled “Evidence of neuroaxonal injury in preeclampsia – a case control study” authors Dr. Bergman et al., presented a case study work where CSF and plasma from 15 women with preeclampsia and 15 women with normal pregnancies were analyzed for four biomarkers S100B, NSE, tau and NfL by Simoa and ELISA. The main finding is women with preeclampsia demonstrated increased CSF and plasma concentrations of NfL which correlated to each other. However, there was no difference in occurrence of either subcortical or periventricular WMLs between the two groups from the MRI findings. The use of Ultra-sensitive Simoa platform is the strong point in this study to measure the very low levels of the biomarkers. The MRI data provided is also adds strength to the study.

The major issue is that the authors ran the samples in just singlicates which needs an attention from the authors. A replicate of two measurement required when running such tests. The conclusion section is overelaborate without through confirmation or evidence of neuroaxonal injury in preeclampsia.

I recommend that this paper for major revisions and also would like to seek response for below comments for further consideration.

Introduction Section:

1. The authors use the term “cerebral biomarkers” (lines 80 and 86) for S100B, NSE, NfL and tau while there are emerging evidence that elevated levels of NfL are a nonspecific marker of neuronal degeneration and do not tell about the underlying cause. I would be cautions of stating them as cerebral biomarkers and rather use the term biomarkers instead of cerebral biomarkers.

We thank the reviewer for this comment. We understand that cerebral biomarkers might not be the perfect expression but in the light of there being so many different biomarkers in preeclampsia we would like to keep the expression cerebral biomarkers since it has also been used in previous work and distinguishing biomarkers reflecting the brain in preeclampsia from biomarkers reflecting angiogenesis and placental dysfunction. If the reviewer prefers another name for this group of biomarkers reflecting cerebral involvement, we would be happy to change to that expression. 

2. In the current section the authors briefly mention about their previous work (lines 80-82). I would like the authors discuss or provide more background or rationale on why they choose the above four biomarker to study preeclampsia. This would keep the readers more engaged to the current work and create a flow.

We thank the reviewer for this comment and have now added a section about the background and the section now reads;

“S100B was the first cerebral biomarker that was reported to be increased in plasma in preeclampsia with severe features by our group and others and in addition, women with visual disturbances demonstrated an increased plasma concentration of S100B. Following these initial reports, there were further studies reporting increased peripheral concentrations also of NSE, NfL and tau, reflecting neurons and axons respectively.”

 (page 5, lines 80-85)

Methods Section:

1. The methods section is confusing. In the lines 120-121 the authors mention that the concentrations of tau and NfL in plasma and CSF were measured using the NF-light and Total Tau 2.0 kit on the Simoa platform. But in the lines 128-130 the CSF Total-tau (T-tau) was measured using the INNOTEST ELISA, and CSF NfL was measured using an in-house ELISA. Is this statement accurate? Why was the CSF T-tau and NfL measured twice using different platforms? Also, what Simoa platform did the authors use (HD1 or HD-X)? Please clarify.

We apologize for this confusion and this has now been corrected in the methods section and it now reads;

“The concentrations of tau and NfL in plasma were measured using the NF-light and Total Tau 2.0 kits using the Simoa HD-1 platform (Quanterix, Billerica, MA) as previously described in detail”

(page 7, lines 123-125)

2. For Simoa analysis the calibrators and quality controls (high and low) are run in duplicates. These are in buffer matrix (the calibrators are in some kind of proprietary buffer from Qaunterix) and usually the duplicate measurements have a %CV of <10-20% (which are acceptable range) on Simoa. One of the serious issue I found is that the authors measured the samples in singlicates (as mentioned in lines 123 and 127). The CSF and plasma are true biofluids and I think a duplicate measurement is required at the minimum to assess the experimental integrity and account for the SD (which is not reported). The % CV is cannot established for any of the samples. This specially applies for the ELISA (less sensitive method) tests where duplicate measurements are required.

We thank the reviewer for this important question. The CVs reported were from two quality control (QC) samples (real plasma samples) that were run in duplicates in the beginning and the end of each run. For NfL, between-run precision was 6.0% at 8.5 pg/mL and 5.1% at 121 pg/mL, while for T-tau, between-run precision was 7.3% at 32.2 pg/mL and 7.0% at 7.5 pg/mL. For ELISA:s, between-run precision was below 10% for all assays. This is now added to the methods and now reads; 

“Two quality control (QC) samples from plasma were run in duplicates in the beginning and the end of each run.”

(page 7, lines 126-127)

and

“Elecsys- and ELISA-based methods were chosen for biomarkers for which Simoa ultrasensitivity was not needed. Between-run precision was <10% for all these assays.”

(page 7, lines 133-135)

3. Why did not the authors include a true control for CSF and plasma? Example: Normal (non-pregnant, age and gender matched CSF and plasma). The reason why I ask is this can serve as a background or endogenous levels for the tested biomarkers in the respective matrix (CSF and plasma). Can the authors comment or show data (either performed by them or any other literature) for the baseline levels of these markers in a non-pregnant women? This experiment is critical to corroborate the current results from this study. This is also the reason why I suggested not to use the term cerebral biomarkers as above (Point 1 of the introduction section).

We thank the reviewer for this comment. In this work we concentrated on pregnant women, in particular since it would have been difficult to obtain CSF from non pregnant women of the same age (women in the study were sampled at the time of caesarian section). Plasma concentrations of S100B and NSE were measured in a longitudinal case control study where non pregnant concentrations remained increased in women with previous preeclampsia compared to previous normotensive pregnancies, although the difference was attenuated compared to during pregnancy (NSE 5.07 vs. 4.28 microg/l, P < 0.05 and S100B 0.07 vs. 0.06 microg/l, P < 0.05).

4. The authors state that all measurements were performed in one round of experiments using a single batch of reagents. While this is a good practice to do, this is also a limited practice as it does not tell about the inter-day variations of the biomarkers tested. Did the authors care to perform an inter-day test to determine the assay variations? This specifically happens with Simoa where the controls (low and high) don’t exactly read the same and Quanterix gives a range (not a specific number). Since the sample size is low and on top the authors ran samples in singlicates, a day to day variation would cause certain degree of changes in the measurements.

We thank the reviewer for the comment about this. Whilst this was not relevant to our study, the inter-day variation for our plasma NfL and T-tau measurements are typically below 10%.

5. I would be interested to look at the Simoa raw data which can be generated from the batch calibration report. Can I request the authors to share this report?

We have attached the requested batch reports from the runs. The batch reports also include some sample matrix comparisons of no relevance to the current study but these were impossible to delete. 

Results Section:

1. While the results are interesting, can the authors comment on why they see (in CSF) increase for some biomarkers (NfL) and decrease for some biomarkers (NSE and tau) compared to the compared to women with normal pregnancies? Though some explanation is offered in the lines 324-327, I did not find this compelling. What does a decrease in biomarker happen in a clinical sense?

We thank the reviewer for pointing this out. We were also surprised by this findings and searching the literature, we don’t find any convincing reason to why this is. The only study conducted previously also showed decreased CSF concentrations of tau. As we wrote, one reason could be reduced neuronal activity. In preeclampsia, there are some findings of impaired cognitive function and on long-term, an increased risk of dementia and stroke. Except for tau, this is the first time these biomarkers are measured in CSF in preeclampsia. We are planning to to similar measurements in CSF and plasma in a population of women with more severe disease (eclamptic fits) to investigate if this pattern holds and what it might depend on. We have added a phrase about this and it now reads as follows;

“There are reports about impaired cognitive function and increased risk of dementia after preeclampsia but if this related to decreased CSF concentrations of tau and NSE remains to be proven.” ……. “These findings need to be confirmed in future studies.”

(page 17, lines 334-336 and 340-341)

2. Also can authors comment on why the S100B change (increase) was noticed in plasma but not serum? And similarly (no difference) for NSE and tau?

We assume that the reviewer means CSF and not serum in relation to plasma. S100B is not only produced in astrocytes but also in extra-cerebral tissues such as muscle cells and adipose tissue. Therefore, the increased peripheral concentrations could be caused by an increased secretion from fat or muscle cells. This has now been stressed and reads as follows;

“Our results showed that S100B was increased in plasma but not in CSF in preeclampsia. This could have different explanations. One reason could be that the signal from S100B produced in other extracerebral tissues such as muscle cells and fat cells are dominating causes to increased plasma concentrations in preeclampsia. Another explanation could be that S100B, produced in astrocytic end-feet close to the blood brain barrier (BBB), is secreted in higher amounts into the blood stream due to BBB injury and thus depleted from the CNS.”

(page 16, lines 310-315)

3. Table 2. Why is the units for S100B and NSE denoted in ug/L, while for NfL and tau the units are in pg/mL? Is there a reason why the authors choose to do so? It would be better to see all units in pg/mL, for uniformity in the table.

We thank the reviewer for this comment. 1 pg/ml = 0.001 ug/L and due to the differences in concentrations we chose to keep these units to make it easier to read. If the reviewer still feels the same units would be better, we could change but then the concentrations of S100B and NSE would be very high. 

4. I am not sure how the hemolysis was observed (line 222) only for seven women in each group. Aren’t the same (aliquot/stock) CSF was used all the other biomarker measurement? In that case all the CSF measurements for other biomarkers (for seven women) should have the hemolysis. Please clarify or am I missing something here?. Also for the lines (304-305)

We apologize for the confusion. The hemolyzed samples were only removed for the NSE analyses since NSE is produced in red blood cells and if NSE would be analyzed in hemolyzed samples, the signal from red blood cells would over-ride any other signal. 

5. I understand that the authors could not perform any stats on the woman with preeclampsia and cerebral edema. But can the authors comment on why the CSF biomarkers (NSE and tau) which was decreased for the preeclampsia group (compared to normal group), is increased for this particular subject? And the same applies for the S100B.

We thank the reviewer for this important comment. As discussed above, it is not yet known why tau and NSE are decreased in CSF in preeclampsia on a group level, though these groups are small and findings need to be repeated in larger sample sizes with different phenotypes of preeclampsia. Our speculation in this case would be that a woman with cerebral edema has a larger cerebral insult and thus a potential increased neuronal, glial and axonal injury. Cerebral edema in preeclampsia is thought to be both vasogenic and sometimes cytotoxic. The cytotoxic edema could potentially be non-reversible, causing persistent injury and this could be the cause to the increased concentrations. Though since this was just one case we chose not to put to much emphasis on that in the discussion section but we look forward to coming analyses of biomarkers in CSF in women with clinical and radiological cerebral complications such as eclampsia and PRES that we are planning for in the coming 6 months. 

Discussion Section:

1. I truly found the lines 286-288 overstated. The results show CSF concentrations of NfL were increased in preeclampsia but there was no evidence presented to indicate any neuro-axonal injury. Are the authors inferring increased NfL in preeclampsia as direct outcome of neuro-axonal injury while the MRI show normal readings? I would re-write this section.

We thank the reviewer for pointing this out. As for MRI imaging, only larger deficits can be detected such as cerebral edema in the acute phase. There are other MRI modalities such as ASL and IVIM to investigate cerebral perfusion and white matter integrity but those were not used in this particular study. NfL is a sensitive marker for axonal degeneration and the fact that this biomarker is increased both in CSF and plasma in preeclampsia, in combination with the known longterm cerebral effects after preeclampsia such as dementia and cognitive deficits strengthens this finding. We have, according to the reviewer’s wish, modified this section a bit and it now reads;

“These findings might indicate a neuro-axonal injury in preeclampsia even when clinical and radiological neurological complications are absent.”

(page 15, lines 293-294)

2. I think the authors should comment and/ or mention in the limitations section on why they could not reproduce earlier findings of increased NSE and tau plasma concentrations in preeclampsia as their older studies (lines 294-297). Similarly the statements made in lines 309-312 can be moved to study limitations rather.

We thank the reviewer for this comment and have moved these sections to ”Limitations” and in addition added a section about the fact that earlier findings could not be reproduced in our population and reads;

“Finally, our findings could not support earlier reports of increased plasma concentrations of tau and NSE in preeclampsia. This could be due to a smaller number of women in our study or other phenotypes of preeclampsia compared to other studies since preeclampsia is a heterogenous disorder.”

(page 18, lines 365-368)

3. A citation is required of the line 298.

Citations are now added.

4. Can the authors support the claim that NfL extensive marker for multiple sclerosis and traumatic brain injury? Is the claim of increased NfL in preeclampsia in the current study is inferred form the above study? Please explain (this goes with the point 1 above).

We have now added references for this statement in the discussion section (ref 29-30)

5. Lines 308-312 there are too much speculation from the authors. I suggest to re-construct this section to tie it to the current study theme.

The section has now been reconstructed and now reads;

“The reason for this is not known. We speculate about that the NSE signal in plasma in this study might be derived from other sources such as red blood cells which might also have impacted previous reports of peripheral concentrations of NSE in preeclampsia.” 

(page 16, lines 318-321)

6. The strength and limitation sections I think is not well written and requires reconstruction. The above limitation (see above) should be incorporated.

The above limitations have now been incorporated into this section as suggested. 

7. Does the MRI examinations reflect a degree of cerebral injury (as stated in lines 347-348)? I don’t see it in Table 3. Please clarify.

We apologize for the confusion. What we meant was that by using MRI, we could characterize the population better in terms of evidence of cerebral edema and white matter lesions. In our population, there were no difference between groups. We have now changed this to;

“the availability of MRI examinations to characterize cerebral involvement”

(page 18, lines 359-360)

Conclusion section:

1. Lines 352-353 is overstated without any direct/tangible evidence for neuroaxonal injury I would refrain from using such sentence without a clear confirmation.

We have now rephrased the section and it now reads;

“We have shown that NfL has potential to be the most promising cerebral biomarker to reflect possible neuroaxonal injury in preeclampsia”

(page 19, lines 371-372)

2. Lines 353-354 is confusing to me. What does the author mean here? Please clarify.

We apologize for the confusion. We realize this phrase is unecessary and it has been removed.

3. Lines 357-358 again is overstated. It is not clear how increased CSF concentrations point to a neuroaxonal injury in preeclampsia and how does detection and facilitate treatment? Does the authors mean increased NfL is a diagnostic marker for preeclampsia?

I sincerely request the authors to reconstruct the whole section without these overstatement. I would appreciate any direct evidence for neuroaxonal injury as the authors state here.

We have rephrased a lot of the conclusion as per above comments and it now reads;

“We have shown that NfL has potential to be the most promising cerebral biomarker to reflect possible neuroaxonal injury in preeclampsia. Though our findings originate from a small study sample and need to be further investigated both in relation to preeclampsia with clinically evident neurological injury such as eclampsia and cerebral edema as well as more subtle neurological impairment in preeclampsia such as neurocognitive deficiencies. Should further studies demonstrate increased NfL concentrations in plasma in cases of preeclampsia with neurological impairment, NfL could potentially have a role as prognostic and /or diagnostic biomarker of cerebral involvement in preeclampsia.” 

(page 19, lines 371-378)

Figure-1

The axis titles of the figure are too small. I would suggest the authors to increase the font size for better resolution.

The axis titles are now increased in font size and also numbers and overall titles.

Reviewer #2: 

This manuscript is a case report study reporting the identification of NFL as biomarker for neuronal injury in preeclampsia. Even though there is a lot of papers reporting on neuronal injury due to preclampsia including papers from the same group, this manuscript is unique in the context of having plasma and CSF samples from the same patient and therefore can correlate the finding between plasma and CSF.

The title is too much strong for the findings and as the authors mentioned in their manuscript that their findings are a pilot study and further analysis are required to confirm their conclusion. the title should be changed

We thank the reviewer for this comment and the title has now been changed to;

“Signs of neuroaxonal injury in preeclampsia – a case control study”

The introduction, materials and methods and result section are very written and presented. however, the discussion need to be readjusted and the finding be discussed more in a global view rather than a point by point discussion. Also. it's clear that the writing style is different from the other parts of the manuscript and re working on the writing style will be more impactful.

We thank the reviewer for these comments and have changed also according to the comments of reviewer 1.

---

## [Decision Letter · Decision Letter 1]

18 Dec 2020

PONE-D-20-29199R1

Signs of neuroaxonal injury in preeclampsia – a case control study

Running head: Neuroaxonal injury in preeclampsia

PLOS ONE

Dear Dr. Bergman,

Thank you for submitting your manuscript to PLOS ONE. After careful consideration, we feel that it has merit but does not fully meet PLOS ONE’s publication criteria as it currently stands. Therefore, we invite you to submit a revised version of the manuscript that addresses the points raised during the review process.

We look forward to receiving your revised manuscript.

Kind regards,

Firas H Kobeissy, PhD

Academic Editor

PLOS ONE

Reviewers' comments:

Reviewer's Responses to Questions

**Comments to the Author**

1. If the authors have adequately addressed your comments raised in a previous round of review and you feel that this manuscript is now acceptable for publication, you may indicate that here to bypass the “Comments to the Author” section, enter your conflict of interest statement in the “Confidential to Editor” section, and submit your "Accept" recommendation.

Reviewer #1: (No Response)

Reviewer #2: All comments have been addressed

2. Is the manuscript technically sound, and do the data support the conclusions?

Reviewer #1: Partly

Reviewer #2: Yes

3. Has the statistical analysis been performed appropriately and rigorously? 

Reviewer #1: N/A

Reviewer #2: Yes

4. Have the authors made all data underlying the findings in their manuscript fully available?

Reviewer #1: Yes

Reviewer #2: Yes

5. Is the manuscript presented in an intelligible fashion and written in standard English?

Reviewer #1: Yes

Reviewer #2: Yes

6. Review Comments to the Author

Reviewer #1: The authors made reasonable edits to the manuscript from the original submission and responded to many of the questions or comments. However few of the questions needs further clarification and response from the authors.

The batch report values for NfL needs attention form the authors for the high values of NfL.

My response and suggestions (in red font) is uploaded as an attachment. Kindly refer to that document.

Reviewer #2: I thanks the authors for their answers to the raised comments and after re-evaluation of the manuscript, i find it suitable for publication.

7. PLOS authors have the option to publish the peer review history of their article (what does this mean?). If published, this will include your full peer review and any attached files.

Reviewer #1: **Yes: **Bharani Thangavelu

Reviewer #2: No

---

## [Author Response · Author response to Decision Letter 1]

4 Jan 2021

Dear Dr Kobeissy, 

Re: PONE-D-20-29199R1; Signs of neuroaxonal injury in preeclampsia – a case control study 

We thank the reviewers for their comments. Here we are pleased to provide our responses. We look forward to further correspondence from PLoS ONE. All changes to the manuscript are marked in yellow.

In addition to the reviewer’s comments, we have also noted an error in the text. Concerning S100B, this has been analysed in serum, not plasma. This has now been corrected throughout the text and highlighted in yellow.

Kind regards,

Dr Lina Bergman, corresponding author

Reviewer 1

The authors made reasonable edits to the manuscript from the original submission and responded to many of the questions or comments. However few of the questions needs further clarification and response from the authors. 

The batch report values for NfL needs attention from the authors for the high values of NfL. 

My response and suggestions as below: 

Introduction

1) Thanks for recognizing that cerebral biomarkers might not be the perfect expression. I would still suggest the authors to use an alternative expression or terminology but try to avoid the term ‘cerebral biomarkers’. 

We thank the reviewer for this comment. As previously mentioned, this term has been used for these biomarkers in other published papers in the area and since it is a term that we need to repeat throughout the manuscript and need a common expression for, we find it hard to exclude it. If the reviewer has other suggestions, we would be happy to look into that. But since preeclampsia is a disease affecting many different organ systems where multiple biomarkers are searched for, such as angiogenic biomarkers, endothelial biomarkers, renal biomarkers etc, we strongly feel we have to provide an expression for biomarkers specifically investigated to reflect cerebral involvement in preeclampsia. 

Methods Section

2) Thanks for the clarification. However my question still remains. Why the samples were only measured in singlicates (Page 7, Line 125)? 

We thank the reviewer for this additional comment. As specified in the manuscript, the analytical variation of the assays is low enough (<10%) to allow for singlicate measurements. Duplicate measurements would not add much and would also result in higher consumption of sample volumes. 

3) Can the authors confirm if the ELISAs were run in singlicates too? If so, this is not common practice for ELISA experiment. 

Whether samples are run as singlicates or duplicates does not depend on the analytical method as such but the precision of the method. The low analytical variation for all the immunoassays allowed us to do singlicate measurements for all analytes. 

4) The two quality control samples used were from plasma (matrix)? Are the authors saying that the controls supplied by Quanterix in the kit are plasma controls? The controls supplied are in buffer matrix. I would suggest the authors to correct the statement w.r.t the controls (these are not plasma controls, refer to Quanterix). 

We are sorry for any confusion. We are saying that we always create our own QC samples to monitor analytical variation across plates and runs in addition to the controls supplied by the kit manufacturer. The precision data presented in the manuscript are data from our own QC samples that were composed of plasma, as stated in the manuscript. 

5) There are no true controls for plasma used in the Simoa assay in the current study. The basal or endogenous levels of these biomarkes are not established. This is why a control (non-pregnant women and age matched) would be highly recommended. 

We thank the reviewer for this comment. As far as we know, no studies have yet published on NfL and tau in the same population with pregnant and non pregnant women. This is beyond the scope of the current study and not important for the research questions addressed here, but we have another manuscript under preparation that deals with this question and that will be published in 2021.

6) Thanks for the response. I understand that CSF source is impossible to get for the age and sex matched subjects. How about plasma samples? I suppose this should be fairly easy source from repositories. My concern is that the endogenous levels for these tested biomarkers in plasma (or blood) will establish a basal levels in non-pregnant women or pregnant women. Measuring the changes or fluctuations in the biomarker levels above or below the basal levels would be interesting to study. Some markers have a basal levels with a range which is very broad. Unless these tested biomarkers are not compared to the basal levels the full scope of the study is very limited. Having said that my question still remains ‘can the authors comment or show data (either performed by them or any other literature) for the baseline levels of these markers in a healthy and non-pregnant women’. 

We thank the reviewer for this additional comment. For S100B and NSE, there are non pregnant data from one year after pregnancy as previously reported. Plasma concentrations of S100B and NSE were measured in a longitudinal case control study where non pregnant concentrations remained increased in women with previous preeclampsia compared to previous normotensive pregnancies, although the difference was attenuated compared to during pregnancy (NSE 5.07 vs. 4.28 microg/l, P < 0.05 and S100B 0.07 vs. 0.06 microg/l, P < 0.05). This paper is cited as number 10 in the reference list for further information. 

In addition, as mentioned above, we are currently analyzing data for all four biomarkers in a cross sectional study comparing women with preeclampsia, normotensive women and non pregnant women but this is yet to be published. Other than that, we have no other repositories of non-pregnant values in the same population / batch of analyses.

7) This question still remains (inter-day variation) specially when the samples are measured in singlicates (Page 7, Line 125) and the controls (in buffer) have a range of values not a specific value (see kit insert from Quanterix, the controls have range of values as determined by the manufacturer). The controls are assigned range due to day to day variations. If the samples were run in replicates of two this question would not have raised. (Please refer above).

The precision data presented in methods cover the total variation of each biomarker at two clinically relevant biomarker concentrations in the relevant sample matrix (plasma) when the samples were analysed (please see Methods). 

8) Thanks for uploading this report (SIMOA raw data). The main issue I found with this report is that the NfL concentration values (pg/mL) of plasma (in fact values for all sample barcode, column 1) are sky high. These several times higher than the highest Calibrator (Calibrator H). Which tells that the measurement of samples needs further dilution and the current values are just the extrapolated values and not the precise values. Also, I am not quite sure if these values in batch report are so high, how can the Figure-1 A have low values? Please clarify. This is a critical concern that the authors need to address. 

We fail to see the reviewer’s concern. The critical parameter to consider is the AEB value; within the range of the calibrator AEBs and none is higher than Calibrator H. Could the reviewer have misinterpreted the report? Please let us know if we could clarify anything further. 

Results section

9) Thanks for adding the above phrase (There are reports about impaired cognitive function and increased risk of dementia after preeclampsia but if this related to decreased CSF concentrations of tau and NSE remains to be proven.” ……. “These findings need to be confirmed in future studies.). I understand that the results are intriguing and the authors observe decrease of above mentioned biomarkers which is hard to explain, but can the authors comment on what does a decrease in biomarker happen in a clinical or medical sense? 

In preeclampsia, it is not clear why tau and and NSE would be decreased in CSF. We refer to the present text in the manuscript that reads;

“An unexpected finding was that women with preeclampsia had reduced CSF concentrations of NSE and tau. In the absence of neurodegeneration, extracellular levels of tau are regulated by neuronal activity-dependent release of the protein, which is likely to determine the corresponding CSF concentrations. It is possible that such a mechanism regulates extracellular NSE concentrations as well. Reduced CSF concentrations of tau and NSE in preeclampsia could thus reflect reduced neuronal activity.”

(page 17, lines 335-340)

We are sorry that we are not able to provide further reflections on why we find these biomarkers in reduced concentrations in the CSF but we hope that future data can elucidate this when investigating more severe forms of the disease and also in relation to neuroimaging findings of cerebral edema.

10) Also can authors comment on why the S100B change (increase) was noticed in plasma but not serum? And similarly (no difference) for NSE and tau?

We assume the reviewer refers to CSF. We have elaborated on this according to below citations;

”Our results showed that S100B was increased in plasma but not in CSF in preeclampsia. This could have different explanations. One reason could be that the signal from S100B produced in other extracerebral tissues such as muscle cells and fat cells are dominating causes to increased plasma concentrations in preeclampsia. Another explanation could be that S100B, produced in astrocytic end-feet close to the blood brain barrier (BBB), is secreted in higher amounts into the blood stream due to BBB injury and thus depleted from the CNS.”

(page 16-17 lines 314-319)

”We speculate about that the NSE signal in plasma in this study might be derived from other sources such as red blood cells which might also have impacted previous reports of peripheral concentrations of NSE in preeclampsia. 

Regarding plasma concentrations of tau, there were no significant differences between groups. In addition, the correlation between plasma concentrations and CSF concentrations of tau in Alzheimer’s disease has proven to be weak and other confounding sources of tau have to be considered. These findings are supported in this study where we did not find any correlation between CSF and plasma concentrations of neither tau, NSE or S100B. Thus, alternative extracerebral sources of tau, NSE and S100B in preeclampsia have to be considered.”

(page 17, lines 325-334)

11) My apologies, I meant CSF. Thanks for adding the above statement. The reasons sound plausible. It would be a good idea to add in the narrative and provide a citation for the basal levels of S100B in normal pregnant women? This goes inline with the above for endogenous levels. 

We have now added this to the manuscript and it now reads;

“Circulating concentrations of S100B in non pregnant women are reported at around 0.06 ug/L.”

(page 17, lines 320-321) 

12) Thanks for making the edits. I would suggest the authors to add the statement about the woman with cerebral edema in the limitations section.

We thank the reviewer for this suggestion and have now modified and added this section which reads as follows;

”It is not yet known why tau and NSE were decreased in CSF in preeclampsia on a group level, though these groups are small and findings need to be repeated in larger sample sizes with different phenotypes of preeclampsia. The only woman with cerebral edema demonstrated increased CSF concentrations of all biomarkers. Our speculation would be that a woman with cerebral edema has a larger cerebral insult and thus a potential increased neuronal, glial and axonal injury. Cerebral edema in preeclampsia is thought to be both vasogenic and sometimes cytotoxic.The cytotoxic edema could potentially be non-reversible, causing persistent injury and this could be the cause to the increased concentrations.”

(page 19, lines 369-376)

Conclusion section:

13) Can I debate that extracerebral tissues (such as muscle cells and fat cells, as stated as a reason by authors as above for increased S100B) may also be acting as a dominating causes to increased plasma concentrations in preeclampsia? Especially since the authors are using ultrasensitive Simoa assays for NfL measurement and also now we can see that the batch report has high NfL (pg/mL) values. Can I get a comment form the authors if my argument is reasonable? 

We thank the reviewer for this comment. Previous literature has mainly reported on median concentrations between group and not relating this to intracerebral engagement. Some studies have reported on correlations to neurological symptoms. Thus, this has been the first study to make an effort to relate the peripheral concentrations of these biomarkers to corresponding concentrations in the CNS through CSF concentrations and also to MRI findings. Since this cohort of women did not experience any clinically evident neurological complications, the interpretation must still be cautios but points towards NfL as the most promising biomarker as mentioned in the main text. We agree with the reviewer that peripheral concentrations of S100B, NSE and tau might be depending on contributions from alternative sources and this was the main aim of the paper to add to previous literature. 

14) I would also suggest the authors to omit the words most and cerebral and instead use ‘NfL has potential to be the promising biomarker’….. 

We are sorry but we don’t agree with the reviewer in this since there are other promising biomarkers for preeclampsia to predict onset of disease and also severity of disease. Though, there are not yet any clinically available objective biomarkers of cerebral involvement in preeclampsia and since preeclampsia is a heterogenous disorders where only a small proportion of women diagnosed will be affected by cerebral complications, there is a need for specific biomarkers for cerebral engagement. If the reviewer wish to replace ”cerebral” with another word reflecting CNS involvement we would be happy to consider to change but we can not only replace it with ”biomarker”.

15) Please refer to above comment or argument and also w.r.t omitting words most and cerebral… 

See response above.

---

## [Decision Letter · Decision Letter 2]

21 Jan 2021

PONE-D-20-29199R2

Signs of neuroaxonal injury in preeclampsia – a case control study

PLOS ONE

Dear Dr. Bergman,

Thank you for submitting your manuscript to PLOS ONE. After careful consideration, we feel that it has merit but does not fully meet PLOS ONE’s publication criteria as it currently stands. Therefore, we invite you to submit a revised version of the manuscript that addresses the points raised during the review process.

We look forward to receiving your revised manuscript.

Kind regards,

Firas H Kobeissy, PhD

Academic Editor

PLOS ONE

Additional Editor Comments (if provided):

Dear Dr. Bergman,

I apologize that the review process took that long, the second reviewer have few minor commenst that I think can be easily addressed. Once answered, the paper is not going through another round of review. I will be accepting it.

Tank you so much and congrats on this elegant work

Reviewers' comments:

Reviewer's Responses to Questions

**Comments to the Author**

1. If the authors have adequately addressed your comments raised in a previous round of review and you feel that this manuscript is now acceptable for publication, you may indicate that here to bypass the “Comments to the Author” section, enter your conflict of interest statement in the “Confidential to Editor” section, and submit your "Accept" recommendation.

Reviewer #1: (No Response)

Reviewer #2: All comments have been addressed

2. Is the manuscript technically sound, and do the data support the conclusions?

Reviewer #1: Partly

Reviewer #2: Yes

3. Has the statistical analysis been performed appropriately and rigorously? 

Reviewer #1: Yes

Reviewer #2: Yes

4. Have the authors made all data underlying the findings in their manuscript fully available?

Reviewer #1: Yes

Reviewer #2: Yes

5. Is the manuscript presented in an intelligible fashion and written in standard English?

Reviewer #1: Yes

Reviewer #2: Yes

6. Review Comments to the Author

Reviewer #1: Thanks for the revised manuscript. However few of the response from the authors is not satisfactory and I am inclining to suggest revisions again.

With regards to the batch report, I offer my apologies for the interpretation w.r.t the calibrators. Perhaps the data format in which it was presented (with the significant digits and a comma instead of period) created the confusion. This confusion is now resolved.

My concerns:

This is a preliminary data and the authors do acknowledge that no studies have yet published on NfL and tau in the same population with pregnant and non-pregnant women. Most of the data is reasonable but the interpretation with NfL is slightly exaggerated. This initial data should be looked at very cautiously as the authors claim that NfL as “most promising cerebral biomarker”. This overemphasis without direct evidence of cerebral damage/involvement and that extra cerebral tissues (muscle cells and fat cells, as stated as a reason by authors for increased S100B) may also be acting as a dominating causes to increased plasma concentrations in preeclampsia. My argument is that other factors could also have affected biomarkers (considering the pregnant conditions) and use of an ultra-high sensitive assay (Simoa). Besides no controls (non-pregnant women) were used and therefore no Simoa NfL data is available from the authors for the range of basal levels of NfL in these age matched non-pregnant women.

Subsequently, several recent studies are emerging to indicate NfL as a concussion biomarker, one should be excessively cautious of the statements used. I think this work is important and should be published, but I think the overemphasis on terms like ‘cerebral’ and ‘most’ promising biomarker without a direct evidence in the manuscript may lead to the results being misinterpreted and misrepresented. As such I emphasize to avoid them. I cannot provide with an alternate expression other than not using the words cerebral and most.

Also, I would like to differ with the authors reasoning for not running duplicate samples. It might be appealing (for whatever reasons) to run each sample in only a single, but running samples in duplicate allows for the calculation of sample variation and provides a measure of the precision of the assay (which cannot be made for the current experiment without inter-day experimental data for the samples). And also the technical error can usually be identified as an outlier well, and that well can be removed from analysis. It is very important to run standards and blanks in duplicate or triplicate to be certain of the precision of these critical measurements. So, I would also suggest the authors to state this under study limitations.

I still feel that this data is valuable and worth reporting, but I think response to above two questions is warranted.

Reviewer #2: (No Response)

7. PLOS authors have the option to publish the peer review history of their article (what does this mean?). If published, this will include your full peer review and any attached files.

Reviewer #1: **Yes: **Bharani Thangavelu

Reviewer #2: No

---

## [Author Response · Author response to Decision Letter 2]

24 Jan 2021

Dear Dr Kobeissy, 

Re: PONE-D-20-29199R2; Evidence of neuroaxonal injury in preeclampsia – a case control study 

We thank the reviewers for their comments. Here we are pleased to provide our responses. We look forward to further correspondence from PLoS ONE. All changes to the manuscript are marked in yellow.

Kind regards,

Dr Lina Bergman, corresponding author

Editor Comments 

Dear Dr. Bergman,

I apologize that the review process took that long, the second reviewer have few minor comments that I think can be easily addressed. Once answered, the paper is not going through another round of review. I will be accepting it. Thank you so much and congrats on this elegant work

We thank the editor for these kind words and are delighted to hear that you appreciate our work.

Reviewer 1

1) Thanks for the revised manuscript. However few of the response from the authors is not satisfactory and I am inclining to suggest revisions again. With regards to the batch report, I offer my apologies for the interpretation w.r.t the calibrators. Perhaps the data format in which it was presented (with the significant digits and a comma instead of period) created the confusion. This confusion is now resolved.

We are happy that the issue is now resolved.

2) This is a preliminary data and the authors do acknowledge that no studies have yet published on NfL and tau in the same population with pregnant and non-pregnant women. Most of the data is reasonable but the interpretation with NfL is slightly exaggerated. This initial data should be looked at very cautiously as the authors claim that NfL as “most promising cerebral biomarker”. This overemphasis without direct evidence of cerebral damage/involvement and that extra cerebral tissues (muscle cells and fat cells, as stated as a reason by authors for increased S100B) may also be acting as a dominating causes to increased plasma concentrations in preeclampsia. My argument is that other factors could also have affected biomarkers (considering the pregnant conditions) and use of an ultra-high sensitive assay (Simoa). Besides no controls (non-pregnant women) were used and therefore no Simoa NfL data is available from the authors for the range of basal levels of NfL in these age matched non-pregnant women.

We thank the reviewer for this comment. Indeed, there might be several different sources of errors for most of the biomarkers used in medicine where none is 100% specific or sensitive for a condition. Though we still argue that there is quite a few published articles on all four biomarkers presented in this article in pregnancy and preeclampsia so this is not the first time to demonstrate increased circulating concentrations of NfL, tau, NSE and S100B in preeclampsia. Though it is the first time, to our knowledge, that data regarding CSF concentrations of these biomarkers is presented in preeclampsia. This was the main purpose of this paper in order to strengthen the cerebral origin and compare peripheral concentrations to CSF concentrations to rule in/rule out if the increased plasma concentrations truly reflected also increased CSF concentrations. This only held true for NfL and in addition, NfL is quite brain specific (https://www.proteinatlas.org/ENSG00000277586-NEFL/tissue). Therefore, we would still like to argue that our findings of increased peripheral NfL concentrations most probably reflects increased secretion of NfL from the brain in preeclampsia. 

We are currently preparing an article about peripheral concentrations of NfL in non pregnant women as a reference group but that is another population and will hopefully be published during 2021. In this study, plasma concentrations of NfL are increased in preeclampsia compared to both normotensive pregnancies and non pregnant women. Even so, we argue that the results from this study can stand alone and that a reference group of normotensive pregnancies is enough in this case. In addition, as argued before, it is almost impossible to obtain CSF samples from healthy non pregnant women of the same age since these samples were obtained at time of spinal anesthesia for a caesarian section. We have put a lot of emphasize in the discussion regarding alternative sources that will always be a confounder to the results.

3) Subsequently, several recent studies are emerging to indicate NfL as a concussion biomarker, one should be excessively cautious of the statements used. I think this work is important and should be published, but I think the overemphasis on terms like ‘cerebral’ and ‘most’ promising biomarker without a direct evidence in the manuscript may lead to the results being misinterpreted and misrepresented. As such I emphasize to avoid them. I cannot provide with an alternate expression other than not using the words cerebral and most.

We have understood that the reviewer would like us to replace the words ”cerebral” and ”most”. Regarding ”cerebral” as explained in earlier responses to reviewers, we can’t find a more suitable expression and in addition, the phrase is commonly used in similar papers describing these biomarkers in preeclampsia. We welcomed an alternative expression but understand that the reviewer also can’t find a more appropriate way of expressing this group of biomarkers. Therefore, we can’t see how we can change this to another phrasing. 

Regarding the expression ”most” we have now, to the reviewer’s request, replaced ”the most” with ”a” in the conclusion in the abstract and in the main text. We think the message in the manuscript is quite clear regarding the need for larger studies of women with clinically evident cerebral complications to confirm these findings. 

Page 4, line 60 and page 19, line 386

4) Also, I would like to differ with the authors reasoning for not running duplicate samples. It might be appealing (for whatever reasons) to run each sample in only a single, but running samples in duplicate allows for the calculation of sample variation and provides a measure of the precision of the assay (which cannot be made for the current experiment without inter-day experimental data for the samples). And also the technical error can usually be identified as an outlier well, and that well can be removed from analysis. It is very important to run standards and blanks in duplicate or triplicate to be certain of the precision of these critical measurements. So, I would also suggest the authors to state this under study limitations. I still feel that this data is valuable and worth reporting, but I think response to above two questions is warranted.

We thank the reviewer for this comment. Though we don’t agree since these analyses are well established and reliable, please see previous comments to the reviewer. Therefore, we think it would be unnecessary to run all of them in duplicates, in particular regarding the small volumes of CSF and the expected outomes of running duplicates (rendering similar results). In contrary, one could argue that if one has a reliable instrument it would not be a reasonable priority to use funds and sample volume to run duplicates if these research funds and biological samples could be used to answer further research questions, using the samples and funds in the most efficient way.

The analytical variation of the assay is monitored using quality control samples as clearly specified in Methods.

---

## [Editor Report · Decision Letter 3]

27 Jan 2021

Signs of neuroaxonal injury in preeclampsia – a case control study

PONE-D-20-29199R3

Dear Dr. Bergman,

We’re pleased to inform you that your manuscript has been judged scientifically suitable for publication and will be formally accepted for publication once it meets all outstanding technical requirements.

Kind regards,

Firas H Kobeissy, PhD

Academic Editor

PLOS ONE
---

## [Editor Report · Acceptance letter]

29 Jan 2021

PONE-D-20-29199R3 

Signs of neuroaxonal injury in preeclampsia – a case control study 

Dear Dr. Bergman:

I'm pleased to inform you that your manuscript has been deemed suitable for publication in PLOS ONE. Congratulations! Your manuscript is now with our production department. 

Kind regards, 

on behalf of

Dr. Firas H Kobeissy 

Academic Editor

PLOS ONE